# A ferroelectric fin diode for robust non-volatile memory

Guangdi Feng[1,2,10], Qiuxiang Zhu[1,2,10], Xuefeng Liu[1], Luqiu Chen[1], Xiaoming Zhao[1], Jianquan Liu[1], Shaobing Xiong[1,3], Kexiang Shan[4], Zhenzhong Yang[1], Qinye Bao[1], Fangyu Yue[1], Hui Peng[1], Rong Huang[1], Xiaodong Tang[1], Jie Jiang[4], Wei Tang[5], Xiaojun Guo[5], Jianlu Wang[6], Anquan Jiang[7], Brahim Dkhil[8], Bobo Tian[1,2] ✉, Junhao Chu[1,3] & Chungang Duan[1,9]

Among today's nonvolatile memories, ferroelectric-based capacitors, tunnel junctions and field-effect transistors (FET) are already industrially integrated and/or intensively investigated to improve their performances. Concurrently, because of the tremendous development of artificial intelligence and big-data issues, there is an urgent need to realize high-density crossbar arrays, a prerequisite for the future of memories and emerging computing algorithms. Here, a two-terminal ferroelectric fin diode (FFD) in which a ferroelectric capacitor and a fin-like semiconductor channel are combined to share both top and bottom electrodes is designed. Such a device not only shows both digital and analog memory functionalities but is also robust and universal as it works using two very different ferroelectric materials. When compared to all current nonvolatile memories, it cumulatively demonstrates an endurance up to $10^{10}$ cycles, an ON/OFF ratio of ~$10^2$, a feature size of 30 nm, an operating energy of ~20 fJ and an operation speed of 100 ns. Beyond these superior performances, the simple two-terminal structure and their self-rectifying ratio of ~ $10^4$ permit to consider them as new electronic building blocks for designing passive crossbar arrays which are crucial for the future in-memory computing.

Ferroelectricity, characterized by a remanent and switchable polarization, just experiences the 100th anniversary of its first discovery in Rochelle salt in 1920[1,2]. The research and application of ferroelectrics were first limited to bulk ceramics, such as in the domains of actuators, piezoelectric transducers, and pyroelectric detectors. Since 1989[3], with the development of thin-films, the switchable polarization by an electric field greater than the coercive field has attracted amounts of effort for the military and commercial applications of ferroelectric films in non-volatile memories[4–15]. The high switching speed at sub-nanosecond and the excellent endurance of over $10^{12}$ make the ferroelectric memory a competitive candidate for replacing the current flash memory products which suffers from slow speed of ~$10^{-3}$ s and

[1]Key Laboratory of Polar Materials and Devices, Ministry of Education, Shanghai Center of Brain-inspired Intelligent Materials and Devices, Department of Electronics, East China Normal University, Shanghai 200241, China. [2]Zhejiang Lab, Hangzhou 310000, China. [3]Institute of Optoelectronics, Fudan University, Shanghai 200433, China. [4]Hunan Key Laboratory of Super Microstructure and Ultrafast Process, School of Physics and Electronics, Central South University, Changsha 410083, China. [5]National Engineering Laboratory of TFT-LCD Materials and Technologies, Department of Electronic Engineering, Shanghai Jiao Tong University, Shanghai 200030, China. [6]Frontier Institute of Chip and System, Fudan University, Shanghai 200433, China. [7]State Key Laboratory of ASIC & System, School of Microelectronics, Fudan University, Shanghai 200433, China. [8]Université Paris-Saclay, CentraleSupélec, CNRS-UMR8580, Laboratoire SPMS, 91190 Gif-sur-Yvette, France. [9]Collaborative Innovation Center of Extreme Optics, Shanxi University, Taiyuan, Shanxi 030006, China. [10]These authors contributed equally: Guangdi Feng, Qiuxiang Zhu. ✉e-mail: bbtian@ee.ecnu.edu.cn

limited endurance of ~$10^4$ cycles[16]. Furthermore, the programmability of polarization renders the ferroelectric memory viable for in-memory computing[17–24], which offers huge advantages in terms of computing power, latency, and energy efficiency by performing parallel multiply-accumulate (MAC) calculations directly with Ohm's law for multiplication and Kirchhoff's law for accumulation[25–28]. Recently, several breakthroughs in materials technology, such as Si-compatible hafnia-based ferroelectrics with 3D integration[29–33], low-weight molecular ferroelectrics[34–36] and polar topology in ferroelectrics[37–40], have further generated unprecedented enthusiasm for the development of ferroelectric memories.

Currently, there are three basic ferroelectric memory structures namely capacitors[3,41], tunnel junctions[42,43] and field-effect transistors[18]. The capacitor-type ferroelectric random access memory (FeRAM) shows outstanding endurance of over $10^{15}$ and is now commercially available in a cell structure of one transistor and one capacitor (1T1C)[41] (Fig. 1a). However, the charge reading of polarization reversal in the capacitor-type FeRAM not only asks for a large area for detectable charges but is also itself destructive and requires a rewrite process after each polarization-reversal reading operation, which hinders the direct analog MAC calculations for in-memory computing. To overcome this destructive readout, the ferroelectric tunnel junction (FTJ) varies its conductance through modulation of the tunnel barrier height by polarization reversal[44,45]. The simple structure of metal/ferroelectric/metal in FTJ allows a high memory density, but the few-nanometers-thick ferroelectric films for direct tunneling suffer from

poor endurance (Fig. 1b)[45]. The alternative three-terminal ferroelectric field-effect transistor (FeFET), where the gate dielectric is replaced by a ferroelectric layer in a standard metal-oxide-semiconductor field-effect transistor (MOSFET), encodes its memory states by modulating the conductance of the channel in the semiconductor by the gate polarization[46]. However, the lack of an epitaxial template results in mesoscopically disordered and polycrystalline ferroelectric on the semiconductor channel which intrinsically leads to uncontrolled device-to-device variation in nanoscale FeFET devices[46] (Fig. 1c). This variation issue makes the commercialization of FeFET challenging. On the other hand, a much simple two-terminal memristor is preferred for a high-density hardware-level in-memory computing[47,48]. Therefore, it is extremely appealing to develop a novel memory technology that not only exploits all merits of fast switching speed, good endurance, non-volatile states, and low operation energy consumption of ferroelectrics, but also combines a simple structure and non-destructive readout mode for high-density memory and computing applications.

In this work, we propose a two-terminal ferroelectric fin diode (FFD) in which a ferroelectric capacitor and a fin-like semiconductor channel are combined to share both top and bottom electrodes (Fig. 1d). This FDD memory absorbs merits of both non-destructive conductance read mode as in FTJs and long endurance as in FeRAM while it allows ferroelectric directly on electrode other than semiconductor, eliminating the intrinsic source of device-to-device variation in a traditional FeFET. Technology computer-aided design (TCAD) simulations demonstrate that the Schottky barrier at the

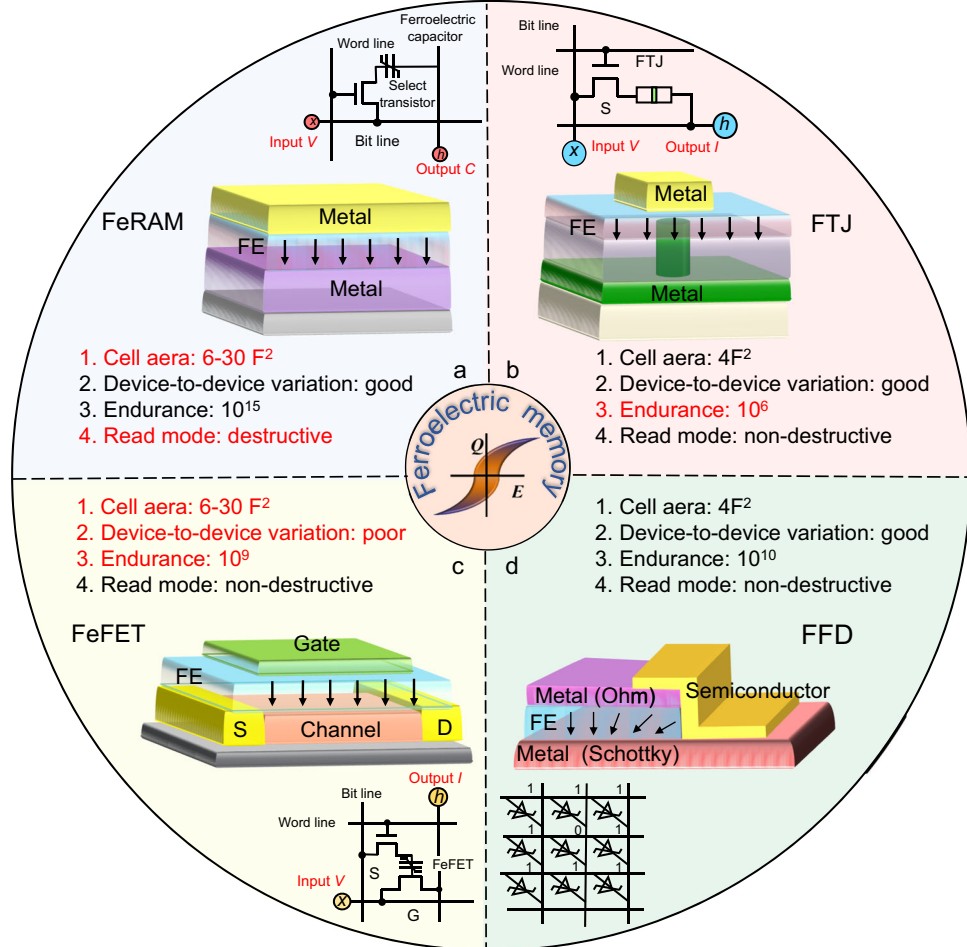

**Fig. 1 | Comparison of ferroelectric memory performances. a–d** Memory performances in the ferroelectric random access memory capacitor (FeRAM) (**a**), ferroelectric tunnel junctions (FTJ) (**b**), ferroelectric field-effect transistor (FeFET) (**c**)

and the proposed ferroelectric fin diode (FFD) (**d**). The parameters in FeRAM, FTJ and FeFET devices are obtained from refs. 41,45,46 respectively.

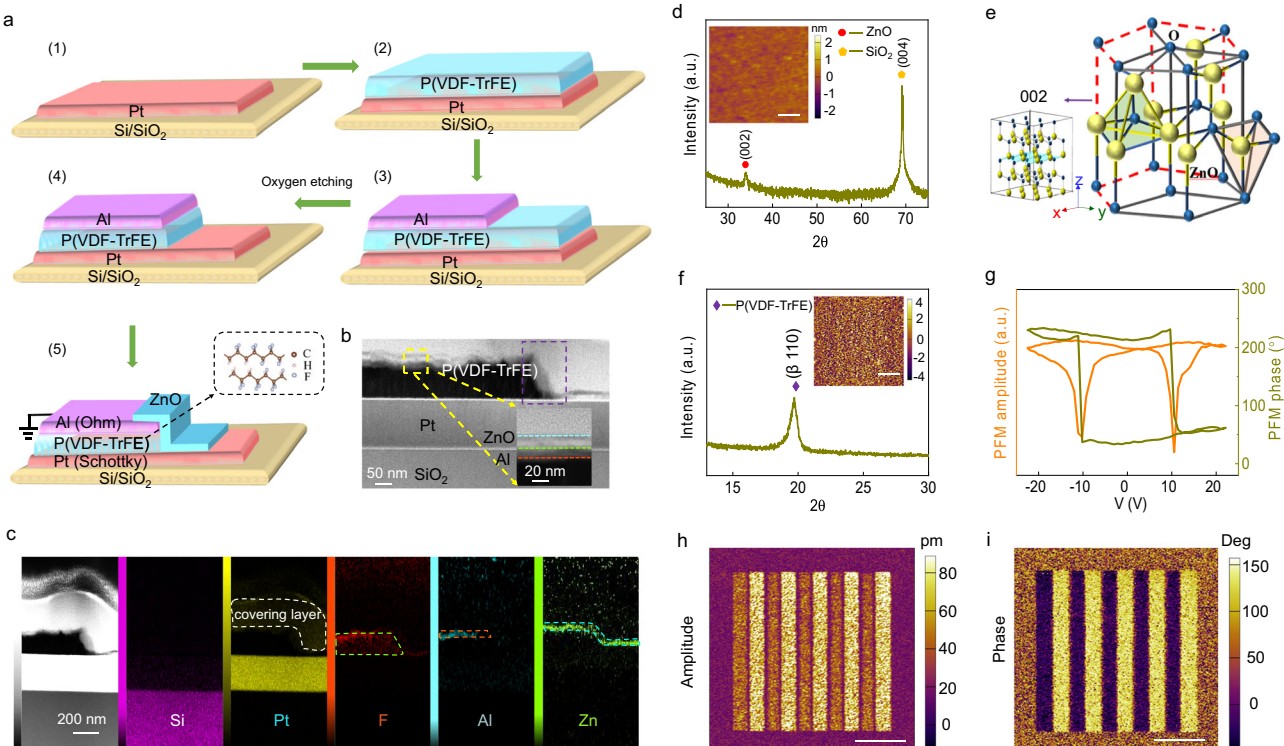

**Fig. 2 | Structure of an FFD device. a** The fabrication process of the FFD based on P(VDF-TrFE). **b** Scanning transmission electron microscopy (STEM) imaging of the device with Pt/ P(VDF-TrFE)/Al/ZnO layers. **c** Energy dispersive X-ray spectroscopy (EDS) mappings of Si, Pt, F, Al, and Zn elements in the device cross profile. **d** The X-Ray Diffraction (XRD) pattern of ZnO films. Inset: The Atomic Force Microscope (AFM) topography image of ZnO films on Pt electrode. Scale bar: 500 nm (**e**) The schematic of ZnO structure with wurtzite hexagonal phase. **f** The XRD pattern of P(VDF-TrFE) films on Pt electrodes. Inset: the AFM topography of P(VDF-TrEF). **g** The butterfly-shaped piezoelectric force microscope (PFM) amplitude loop and 180°-reversed PFM phase loop of P(VDF-TrFE) films on Pt electrodes. **h, i** The PFM amplitude image (**h**) and phase image (**i**) after scanning the grounded PFM tip on the P(VDF-TrFE) films with a + 20 V and −20 V bias alternately applied to the Pt electrode. Scale bar in (**f, h, i**) has the same value of 3 μm.

semiconductor/electrode interface twists the electric field distribution, adding a transverse electric field to the vertical semiconductor channel of the FFD. The remanent polarization plastically-reversible by the distorted electric field permits the FFD both digital and analog memory functionalities. Such a FFD operates using different ferroelectric materials (namely the organic P(VDF-TrFE) polymer and inorganic industrially used PZT compounds) which emphasizes its universal character. Compared to the vast family of the current nonvolatile memories, this FFD shows exhaustive properties of prior memories with high performances such as an endurance of over $10^{10}$ cycles, an ON/OFF ratio of ~$10^2$, a feature size of 30 nm, an operating energy of ~ 20 fJ and an operation speed of 100 ns. Benefiting from the simplicity for fabricating this FFD and its self-rectification ratio of ~ $10^4$, a passive crossbar array with 1.6 k units is constituted and used to demonstrate the in-memory computing of a simple pattern classification task. The high device-to-device uniformity is reflected by a small σ/μ value of ~0.023 for positive coercive voltage and ~0.019 for negative coercive voltage using a Gaussian distribution. This work opens a new avenue for efficient memories and emerging-computing architectures for big data and artificial intelligence applications.

## Results
### Structure of two-terminal FFD
Ferroelectric thin films sandwiched between a top electrode and a bottom electrode are used to support a sidewall semiconductor, forming a vertical fin-like structure where the channel length defined by the thickness of the ferroelectric can be easily controlled at the nanoscale. As will be discussed below, by inducing a Schottky contact, for example between the bottom electrode and the fin-like

semiconductor channel/bridge, the bottom electrode then plays a dual role i.e.; both the drain and the gate of a typical FeFET. While the fin-like semiconductor channel contributes most currents, the ferroelectric domain switching rearranges the electric field configuration at ferroelectric/semiconductor interface and results in a resistive switching. This combination structure of sandwiched ferroelectric and fin-like Schottky sidewall is morphologically called as ferroelectric fin diode (FFD).

The FFD device can be easily fabricated using a common photolithography technique. Figure 2a presents the fabrication process of a FFD with the ferroelectric P(VDF-TrFE) and a n-type ZnO channel (please refer to the Methods section for details). The final heterostructure is examined by cross-sectional scanning transmission electron microscopy (STEM) (Fig. 2b, c) that shows a clear stair-like device structure. According to the images of the energy dispersive X-ray spectrometry (EDS) mapping of Si, Pt, F, Al, and Zn elements, the expected FFD device schematized in Fig. 2a5 is indeed confirmed. The thickness of Pt/P(VDF-TrFE)/Al/ZnO layers is 150/100/20/30 nm, respectively. Note that the oblique ZnO channel with a deflection angle of 30° to the vertical may arise from the nonlinear etching process. Since the ZnO semiconductor and the P(VDF-TrFE) ferroelectric layers play an important role in this novel device, their topography and crystal structure are checked by atomic force microscope (AFM) and X-ray diffraction (XRD), respectively. Deposited on Pt electrode using magnetron sputtering method, the ZnO films show AFM images (inset of Fig. 2d) with no macroscopic defects, pinholes or visible cracks and a root mean square roughness of 0.52 nm, which suggests that uniform and smooth ZnO films are obtained. The P(VDF-TrFE) films on Pt electrode have also a good quality and uniformity with a root mean square roughness of 2.01 nm

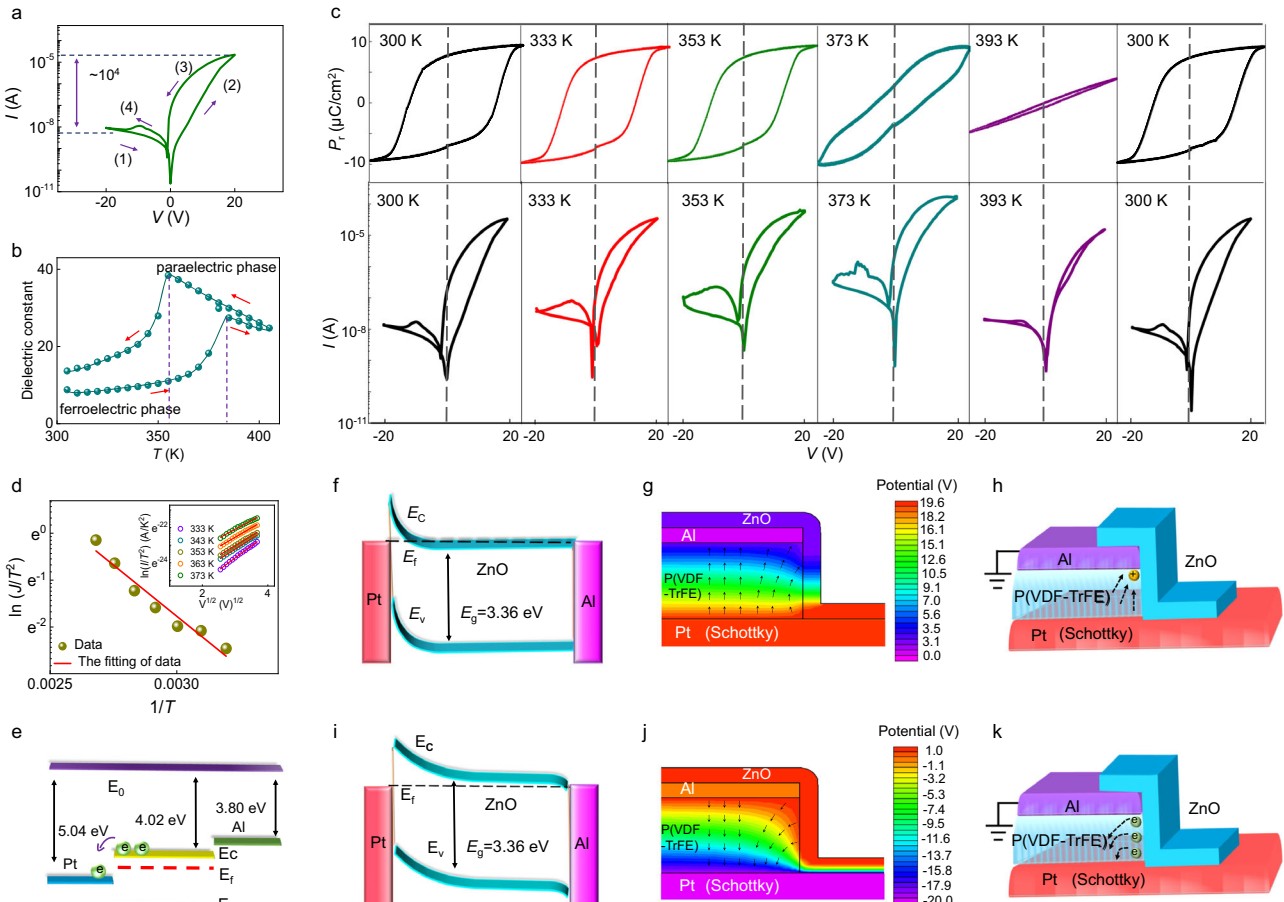

**Fig. 3 | Polarization-driven resistive switching. a** The quasi-static current versus voltage (*I–V*) curve in a typical FFD. **b** The temperature-dependence of the dielectric permittivity. **c** The polarization versus voltage *(P-V)* curves at a frequency of 100 Hz and quasi-static *I–V* curves of the FFD at the same temperatures of 300 K, 333 K, 353 K, 373 K, 393 K and back to 300 K, respectively. **d** ln(J/T²) versus 1/T plots at the voltage bias of 5 V. The linear fit gives a Schottky barrier height of ~0.85 eV. Inset: ln(J/T²) versus V^(1/2) plots at temperature of 333 K, 343 K, 353 K, 363 K, 373 K, respectively. **e** The schematic diagram of the electronic band structure of Pt electrode, ZnO semiconductor and Al electrode. **f–k** The TCAD-obtained energy band alignment of Pt/ZnO/Al structure after poling by a 20 V bias (**f**) and −20 V bias (**i**). The TCAD-obtained distribution of electric potential in the FFD under poling of 20 V bias (**g**) and −20 V bias (**j**). The scheme of polarization vectors alignment in the FFD after poling by 20 V bias (**h**) and −20 V bias (**k**).

(inset of Fig. 2f). The XRD patterns show a Bragg peak at 34.1° for ZnO (Fig. 2d) and 19.6° for P(VDF-TrFE) films (Fig. 2f), which are specific to the wurtzite hexagonal phase (Fig. 2e) and ferroelectric β phase[49], respectively. The ferroelectricity of P(VDF-TrFE) films is further confirmed by the butterfly-shaped loop of the piezoelectric force microscope (PFM) amplitude and 180°-reversed loop of the PFM phase (Fig. 2g). The reproducible ferroelectric domain switching is attested by the out-of-plane PFM images showing alternate of up and down polarization domains. Indeed, in Fig. 2h, i, stripe domains characteristics of ferroelectricity are written by alternately applying a +20 V and −20 V bias to the Pt electrode while the PFM tip is grounded. A clear 180° phase shift is observed between adjacent domains (Fig. 2i) with depressed amplitude value at domain walls (Fig. 2h).

**Polarization-driven resistive switching**

By quasi-statically sweeping a voltage from −20 V to +20 V on the bottom Pt electrode, the semi-logarithmic current versus voltage (*I–V*) curve shows an anticlockwise hysteresis with an ON/OFF ratio of ~10² (Fig. 3a). Moreover, the *I–V* curve of the FFD further demonstrates that the structure of this new device can significantly reduce the reverse (*V* < 0) current (Fig. S1). To demonstrate whether the observed resistive switching is dominated by the polarization reversal, ferroelectric-phase-dependent resistive switching experiments are performed. As shown in the temperature-dependent dielectric

permittivity of our P(VDF-TrFE) films (Fig. 3b), the ferroelectric-to-paraelectric phase transition occurs at a Curie point $T_c$ of 380 K on heating while it drops to $T_c = 355$ K on cooling, which agrees well with previous reports[50]. This thermal hysteresis of 25 K indicates that the ferroelectric transition of this copolymer is of first order. This phase transition is further attested by directly measuring the polarization versus voltage (*P–V*) loops. As shown in top panels in Fig. 3c, *P–V* loops, measured at 100 Hz, exhibit clear hysteresis with a remanent polarization (*P_r*) of ~7.5 μC/cm² at room temperature. As the temperature increases, the *P–V* loop is altered at 373 K (both *P_r* and the coercive voltage *V_c* reduce) and disappears at 393 K i.e., above $T_c = 380$ K evidenced on heating with temperature dependent dielectric permittivity, resulting in a linear behavior characteristic of a paraelectric state. When cooling back to 300 K, the *P–V* hysteresis loop recovers. Likely, the resistive switching (bottom panels in Fig. 3c) behaves concomitantly with the *P–V* loop and vanishes when the P(VDF-TrFE) films are in the paraelectric phase. Note that the operating voltage can be significantly decreased by lowering the thickness of the ferroelectric layer and/or selecting ferroelectric materials with small coercive fields (Fig. S1b–f) to meet the CMOS technology requirements. Note that a tradeoff issue between operation voltage and conduction current, that is, a thinner film for lower operation voltage results in higher current, is usually suffered in 2-terminal resistive devices. In this view, it will be more energy efficient to decrease the operation voltage by using

ferroelectric materials that possesses a much small coercive field. The direct correlation between ferroelectricity and resistive switching strongly supports that the resistive switching in the FFD device is dominated by the polarization reversal.

To eliminate the influence of other possible mechanism such as defects or charging effect, we fabricate a referenced device sample in which the ferroelectric layer is replaced by a non-ferroelectric aluminum oxide ($Al_2O_3$) (Fig. S2a). The thickness of $Al_2O_3$ and ZnO layer in the referenced device is 27 nm and 30 nm respectively (Fig. S3). The resistive switching in the FFD devices shows counterclockwise $I–V$ curves in the first quadrant. However, when ferroelectric layer is replaced by dielectric of $Al_2O_3$, a small clockwise $I–V$ curve is observed (Fig. S2b). This clockwise hysteresis can be understood by the charge trapping effect[51] or the presence of impurities in the ZnO or interface. The opposite hysteresis in ferroelectric device with that in $Al_2O_3$ dielectric confirms that the counterclockwise hysteresis in our ferroelectric device is not dominated by the charge trapping effect or impurities effect. A FeFET device is then fabricated and shows that the n-type ZnO channel can be effectively tuned by ferroelectric polarization (Fig. S4), implying that the charge trapping or impurities effect are much weak and negligible compared with the field effect by ferroelectric polarization.

It is worth mentioning that the $I–V$ curves are asymmetric with a self-rectifying ratio of ~ $10^4$ (Fig. 3a). This self-rectifying characteristic results from the Schottky contact between the Pt electrode and n-type ZnO films. The work function of Al, Cu, Au and Pt metals is obtained from ultraviolet photoelectron spectroscopy (UPS) to be 3.80 eV, 4.63 eV, 4.86 eV and 5.04 eV respectively, while an affinity of 4.02 eV is obtained in ZnO semiconductor (Fig. S5). With a clean interface contact, a higher work function than the affinity enables a Schottky contact at metal/ZnO interface, while a lower work function results in an ohmic one. The Pt (5.04 eV)/ZnO (4.02 eV) Schottky barrier and Al (3.80 eV)/ZnO (4.02 eV) ohmic contact are confirmed by their rectified and liner $I–V$ curves respectively (Fig. S6). The Schottky barrier of the Pt/ZnO interface is further confirmed by the linear relationship between $\ln(I/T^2)$ and $V^{1/2}$ (Fig. 3d). By measuring the temperature-dependent slope of this linear relationship, a barrier height value of ~ 0.85 eV is obtained and agrees well with the energy band alignment when considering the energy level of the ZnO channel and Pt and Al electrodes (Fig. 3e).

To better understand how the polarization affects the electronic transport, a TCAD simulation is performed. Interestingly, because of the existence of Pt/ZnO Schottky barrier, there is lateral electric field pointing forward (backward) to the vertical ZnO channel when a voltage of +20 V (-20 V) is applied to the Pt electrode (Fig. 3g and j). For example, when a voltage of -20 V is applied to the Pt electrode, the negatively-biased Schottky barrier suffers most voltage and results in curved electric potential in the ferroelectric layer (Fig. 3j). The electric field, defined by minus the gradient of the electric potential, is represented with black arrows in Fig. 3g and j. The lateral electric field, estimated to be of ~ 140 MV/m (near the Al/ZnO interface) or -200 MV/m (at most ZnO channel region) under the +20 V or -20 V bias, is high enough to reverse the polarization in ferroelectric copolymers[21,22]. Figure 3h, k show the phenomenological model of polarization state after removing the voltage bias. By sweeping a negative voltage that is larger than the coercive voltage (-$V_c$ ~ -15 V) on the Pt electrode, the lateral electric field makes the polarization obliquely backward to the vertical ZnO channel. The remanent polarization with net negative bound charges at the ferroelectric/channel interface electrically weakens the carrier density of the n-type channel[12,17] and leads to a thicker Schottky barrier (Fig. 3i), resulting in the low conductance state (LCS). These bound charges can be switched away by aligning the polarization upward, even forward to the vertical ZnO channel near the Al/ZnO interface, when applying a positive voltage higher than +$V_c$ (Fig. 3h), which permits the Schottky barrier to recover its original

state (Fig. 3f) and gives the high conductance state (HCS). This model explains well the observed asymmetric resistive switching (Fig. 3a) because of the non-uniform and amplitude-dependent lateral field in the ferroelectric layer (Fig. 3g and j) due to the vertical ZnO channel in FFD devices. This contrasts with the symmetric behavior in more traditional FeFET (Fig. S4) with expectedly vertical uniform field.

## Robustness performance and universality of FFD

The fatigue behavior is a very important parameter, which is related to the service life of memory devices and the prospect of their potential industrialization. The ferroelectric and electrical fatigue characteristics of FFDs have been studied systematically. Figure 4a, b (up panels) and S7a–d demonstrate the performance of a typical P(VDF-TrFE)-based FFD. The remanent polarization ($P_r^+$ and $P_r^-$) and the corresponding $P–V$ loops as a function of measured cycles are presented in Fig. 4a (up panel) and Fig. S7c respectively. Interestingly, accompanying with the robust polarization reversal of up to $10^7$ cycles, the device shows stable resistive switching throughout (Fig. 4b (up panel) and Fig. S7d). It is reasonable that the conductance sustains a better endurance than $P_r$. The contribution to the resistive switching significantly involves the lateral region around the ferroelectric/semiconductor interface where the up-left switching (Fig. 3g, h and j, k) is more robust as it does not involve nucleation like in the up-down switching in the parallel capacitor region between bottom Pt electrode and top Al electrode, where $P_r$ is measured.

To highlight the universal aspect of both these robust performances and the device structure itself, an industrialized inorganic ferroelectric, i.e. lead zirconate titanate (PZT) on a 4-inch Si wafer (Fig. S8), is used to fabricate the FFD (Fig. S9) and put in parallel to the one made with the organic P(VDF-TrFE) ferroelectric. As expected, an endurance of up to $10^{10}$ cycles in both polarization reversal and resistive switching is obtained in a typical PZT-based FFD (Fig. 4a, b (bottom panels) and S7e, f). The same self-rectifying and counterclockwise characteristics of the resistive hysteresis in such devices based on both inorganic PZT and organic P(VDF-TrFE) indicates that the as-designed FFDs are already at high technology readiness level.

Figure 4c shows the retention property of FFDs. Benefiting from the nonvolatility of ferroelectrics, both HCS and LCS show slight degradation, if any, in a period of ~$10^4$ s in either P(VDF-TrFE) (top panel) or PZT (bottom panel) based devices. In addition, the programming speed of FFDs is studied (Fig. S10a, b). The device is firstly written to LCS by applying a -28 V (-8 V) pulse with a width as long as 100 ms to the P(VDF-TrFE) (PZT) device. Then +35 V ( + 10 V) pulses with an increased width from 1 μs to 1 ms (100 ns to 100 μs) are used to write the P(VDF-TrFE) (PZT) device to HCS. Each programing voltage is followed by a read voltage pulse of +3.0 V (+0.2 V) with a width of 100 ms to check the conductance state. The achieved speed of P(VDF-TrFE)- and PZT-based devices is found to be 1 μs and 100 ns, respectively.

The essential width of Al top electrode needed for the resistive properties is the limit of feature size in our devices. A reversed structure is used to verify the scalability of our memory devices (Fig. S11). This reversed structure allows the design of across (I in Fig. S11d, e), just touch (inset of Fig. 4d and II in Fig. S11d, e) and separate (III in Fig. S11d, e) between vertical projection of Al electrode and Pt electrode where the just touching case is equivalent to infinitely reducing the width of Al. It is found that the resistive switching exists well when Al electrode and Pt electrode have across (Fig. S11f) and even just "zero" overlap (Fig. 4d) within our photolithography error, implying a nanoscale scalability in our memory devices. Inspired by this, we further fabricated nano devices where the width of Al top electrode is only 30 nm (Fig. 4e and Fig. S12). As shown in Fig. 4f, the resistive switching survives well in the 30-nm nano device.

The energy consumption for each write operation of the ferroelectric fin diode is evaluated using the formula: $E = UIt$. For example, when an operating voltage with an amplitude of 20 V is used

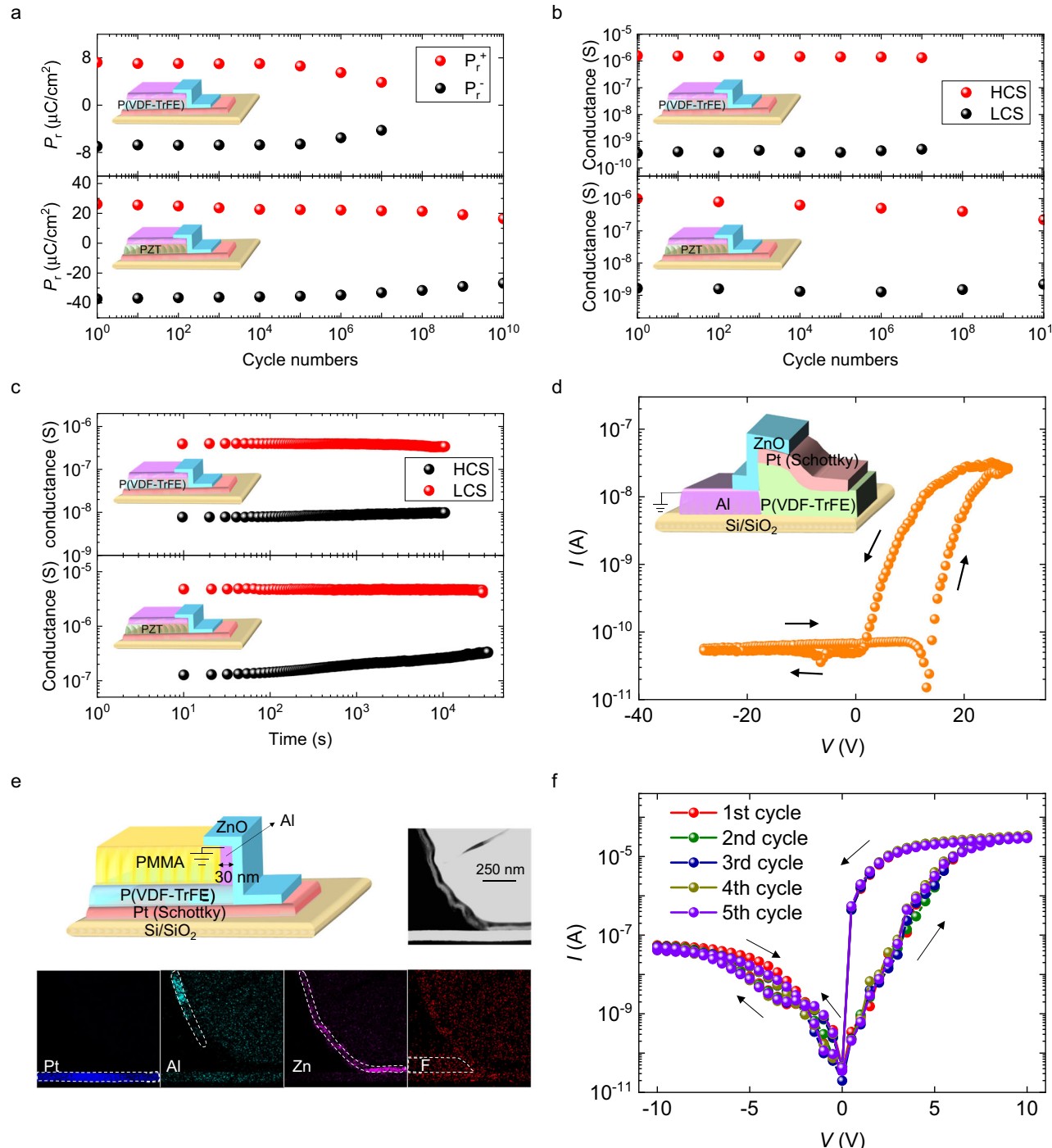

**Fig. 4 | Robust memory cycling and universality. a**, **b** The evolution of the remanent polarization ($P_r$) (**a**) and conductance (**b**) with endurance cycles for a typical FFD based on P(VDF-TrFE) (top panels) and PZT (bottom panels). **c** The retention characteristics for a typical FFD based on P(VDF-TrFE) (top panel) and PZT (bottom panel). Insets give the scheme of the device structure. **d** The quasi-static *I–V* curves of a reversed FFD device with just "zero" overlapped electrode pairs of Al and Pt. During the electrical measurements, the Al electrode is grounded as in other devices. **e** STEM imaging of a FFD nano device where the width of Al top electrode is only 30 nm and the EDS mappings of Pt, Al, Zn and F elements in the device cross profile. Inset shows the schematic diagram of the FFD nano device. **f** The quasi-static *I–V* curves of a typical FFD nano device.

for a reversed FFD based on P(VDF-TrFE) (Fig. S11f), the energy consumption is estimated to be ~1 fJ and ~ 20 fJ for a reset operation and a set operation respectively.

**Device uniformity and analog storage for in-memory computing**

In a traditional ferroelectric transistor (Fig. 1c), the ferroelectric layer has to be deposited on the semiconductor layer other than a seed electrode.

The lack of an epitaxial template results in poor ferroelectric quality that is usually mesoscopically disordered and polycrystalline[46]. This leads to uncontrolled device-to-device variation in nanoscale devices. On the contrary, in a FFD (Fig. 1d), the ferroelectric layer is directly deposited on a seed electrode (for example, Pt electrode for PZT films) and covered by a top electrode. This sandwiching metal/ferroelectric/metal structure, indeed the same as that of a commercial ferroelectric capacitor,

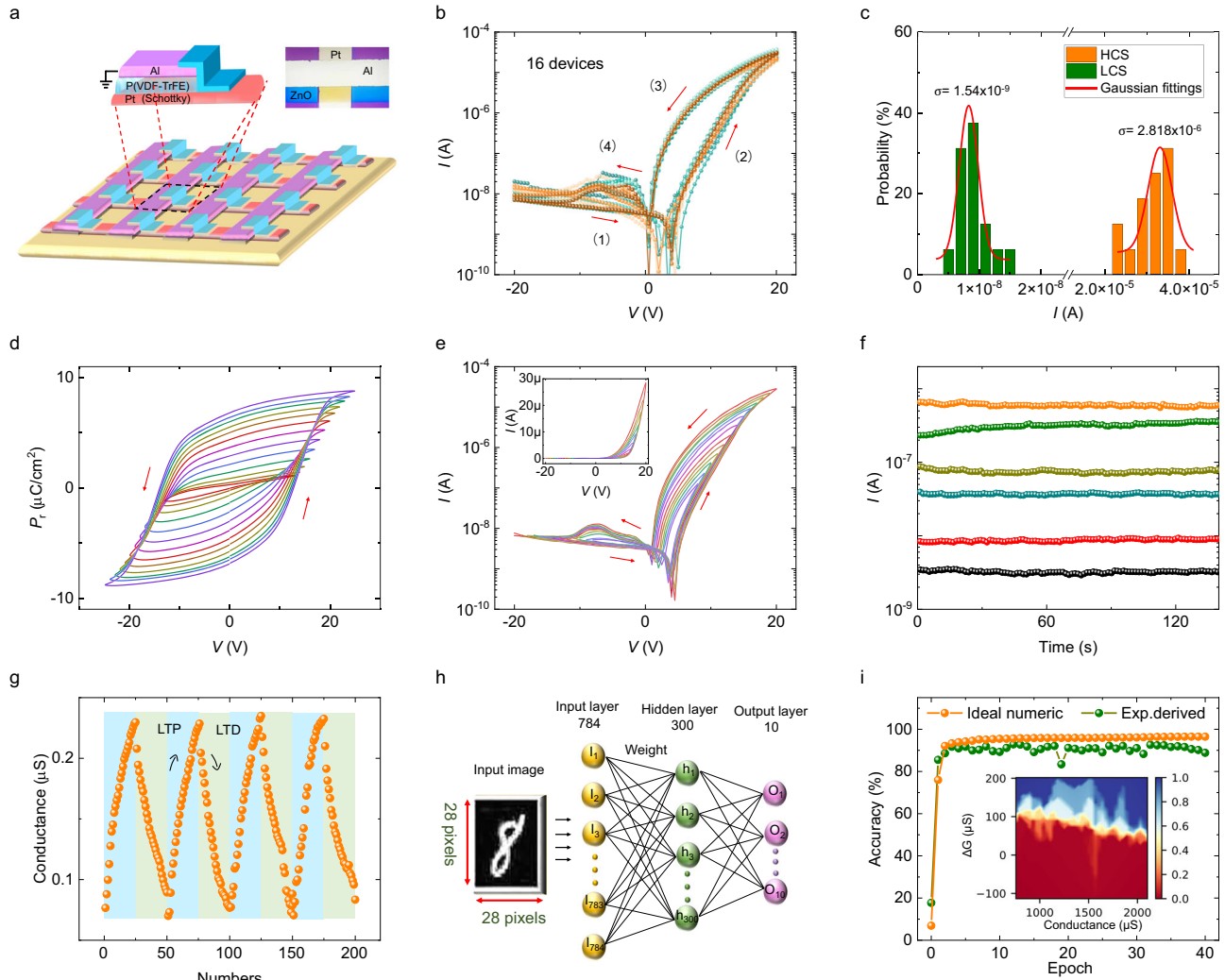

**Fig. 5 | The uniformity and analog storage for in-memory computing. a** The schematic diagram of a 4 × 4 passive crossbar array constituted by FFDs on a Si/SiO₂ substrate. Insets show the schemes (left) and the optical image (right) of a unit device. **b** The overlay plots of quasi-static I–V curves of all 16 devices. **c** The distribution of LCS and HCS conductance states for all 16 devices. The red lines indicate the fit by gaussian distribution. **d** The multiple P–V curves of the FFD based on P(VDF-TrFE) with a constant sweeping frequency of 100 Hz and a continuously increased amplitude from 13 V to 25 V with a step of 1 V. **e** The multiple quasi-static I–V curves in a logarithmic coordinate obtained by gradually changing the sweeping amplitude from 8 V to 20 V with a step of 1 V. Inset gives the I–V curves in a linear coordinate. **f** The retention characteristics of six intermediate conductance states. **g** The evolution of conductance potentiation and depression (i.e. LTP and LTD) during the poling with a repeated voltage pulse sequence made by 25 positive voltage pulses (+25 V/10 μs) followed by 25 negative voltage pulses (−18 V/10 μs). A read voltage pulse (+3 V/10 μs) follows each poling voltage pulse to obtain the nonvolatile conductance. **h** A schematic diagram of a three-layer artificial neural network. **i** Evolution of the accuracy with training epochs achieved by simulating the FFDs-based artificial neural network for recognizing handwritten digits with 8 × 8 pixels. Insets show the probability distributions of the change in conductance (ΔG) induced by a write operation versus initial conductance at potentiation process.

possesses a high ferroelectric quality and good uniformity even in nanoscale. Note that during the following semiconductor deposition, the ferroelectric layer is protected by the top electrode. Thus, a good uniformity is expected in ferroelectric fin diodes.

A 4 × 4 passive array with P(VDF-TrFE)-based fin diodes (Fig. 5a) is fabricated on a Si/SiO₂ (300 nm thick) wafer. The current cross-talk issue in this passive crossbar array can be effectively eliminated by the self-rectifying characteristic in each device unit[48]. The device uniformity is checked by performing I–V curves in all 16 device units (Fig. 5b) which shows that the dispersion of the response from one device unit to another is very small. The device-to-device variation is evaluated using the ratio of σ/μ in a Gaussian distribution function $f(G) = \frac{1}{\sqrt{2\pi}\sigma} e^{-\frac{(G-\mu)^2}{2\sigma^2}}$, where μ and σ are the mean value and standard deviation of the current, respectively. A good uniformity is found with a σ/μ value of -0.18 for HCS and -0.08 for LCS, respectively (Fig. 5c).

The electronic transport of a typical device unit is comprehensively measured. Five successive I–V curves between -20 V and +20 V are collected and show little deviation (Fig. S13). Ferroelectric is well known for its nonvolatility and analog programing characteristic[52]. Figure 5d demonstrates the analog switching of polarization with multiple $P_r$ in amplitude-variant P–V curves of the FFD based on P(VDF-TrFE). Accordingly, by changing the sweeping amplitude of the positive voltage from 8 V to 20 V with a step of 1 V, clearly separated and nested hysteresis loops are observed (Fig. 5e), implying multiple intermediate conductance states in the FFD. This is caused by the multiple intermediate polarization states in the ferroelectric layer as shown in Fig. 5d. The nonvolatility of these intermediate conductance states can be reflected by these open hysteresis loops. Six distinguishing states are also chosen to measure their retention characteristic and no degeneration is observed in a time scale of >100 s for all six states (Fig. 5f). These persistent multiple conductance states provide

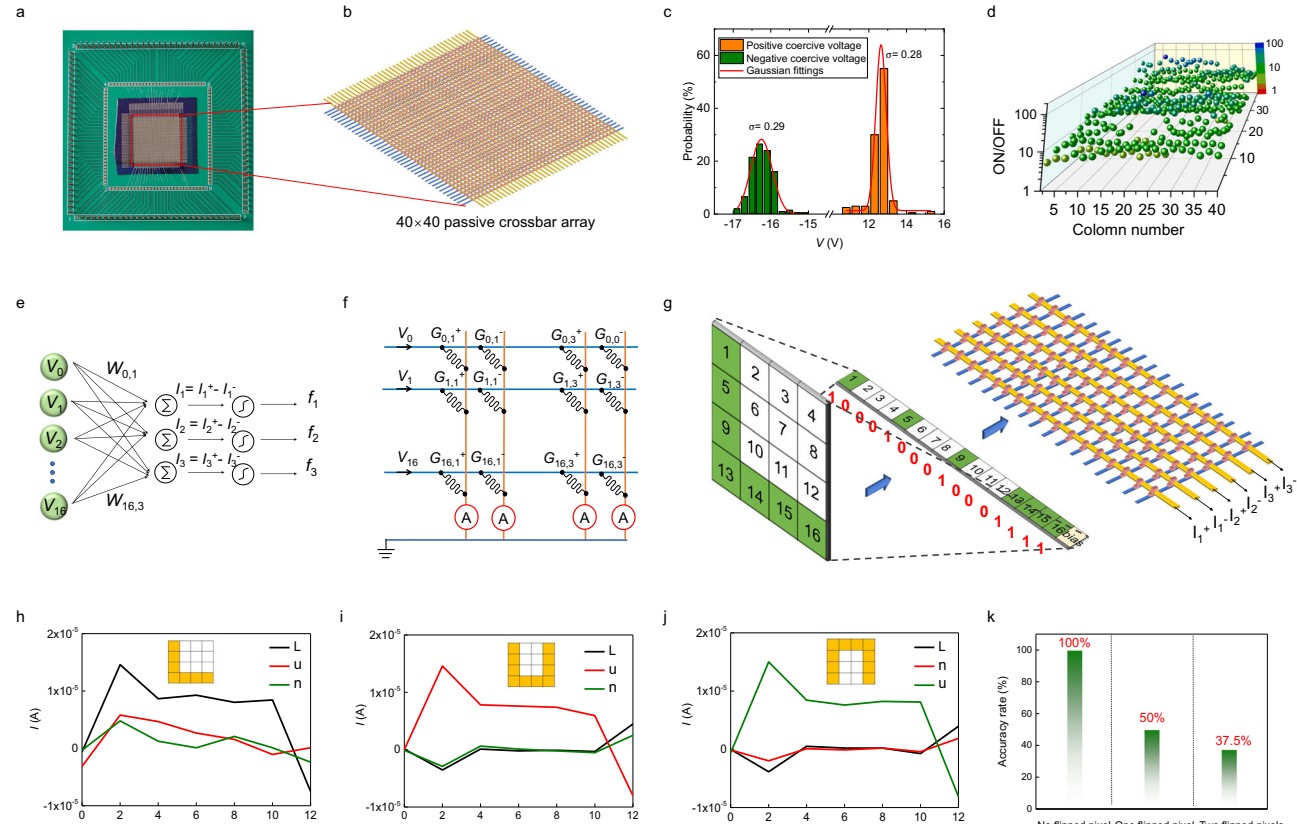

**Fig. 6 | In-memory computing of a pattern classification task within a passive crossbar array. a, b** The optical image (**a**) and schematic diagram (**b**) of a 40 × 40 passive crossbar array based on FFD devices. **c** The distribution of negative coercive voltage and positive coercive voltage obtained from transient *I–V* curves in Fig. S16. The red lines indicate the fit by gaussian distribution. **d** Statistics of resistive switching ratio at a read voltage of 3 V in the 40 × 40 passive crossbar array. **e** A schematic diagram of an artificial neural network (ANN) for clarifying three 4 × 4-pixel images. **f** The schematic diagram indicating how the hardware to implement

the ANN in (**e**). **g** The schematic diagram indicating how the image are encoded to input into the hardware ANN. Each weight is encoded by conductance difference between neighboring pair units. **h–j** The current difference ($I_i = I_i^+ - I_i^-$) collected from columns when images of "L" (**h**) "u" (**i**) and "n" (**j**) are inputted into the 16 × 6 hardware ANN. The patterns recognition is implemented with the value of $I_i$ standing for the target *i* image being the biggest. **k** When zero, one, and two pixels are randomly flipped, and recognition accuracy is 100%, 50%, and 37.5%, respectively.

the essential ingredients to emulate the so-called synaptic plasticity, plastic changes of synaptic weights, that is at the heart of the learning processes. The quasi-linear conductance potentiation (strengthening) and depression (weakening) with 25 discrete states can be achieved (Fig. 5g) by repeatedly applying a voltage pulse sequence of 25 positive voltage pulses (+25 V/10 μs) followed by 25 negative voltage pulses (-18 V/10 μs), respectively (Fig. S14). This analog behavior of conductance resembles the long-term potentiation/depression (LTP/D) process in synaptic devices.

An artificial neural network constituted by crossbar array based on FFDs for digits recognition is simulated using the experimentally measured conductance states. Both the small image version (8 × 8 pixels) of handwritten digits from the "Optical Recognition of Hand-written Digits" (ORHD) dataset and the large image version (28 × 28 pixels) of hand-written digits from the "Modified National Institute of Standards and Technology" (MNIST) dataset are used to perform the back-propagation simulation. For the large (small) image version, a multilayer perceptron (MLP) neural network with 784 (64) input neurons, 300 (30) hidden neurons, and 10 output neurons is utilized by using Cross-Sim simulator (Fig. 5h)[53]. In the simulation, the crossbar, regarded as part of a "neural core", performs vector-matrix multiplication and outer-product update operations. Generally, the performance of a neural network is greatly influenced by the controllability (e.g., nonlinearity and write noise) of synaptic devices, which can be quantitatively analyzed using the probability distribution

of the conductance change ($\Delta G$) induced by a write operation. The plots of $\Delta G$ versus initial $\Delta G_O$, derived from the cyclic conductance potentiation and depression, are presented in insets of Fig. 5i and Fig. S15, respectively. For small digits, the classification accuracy approaches 82.6% within the second training epoch and approaches 92.6% after 16 training epochs, which is close to 96.4% the theoretical limit of an ideal numeric training for small digits (Fig. 5i). For large digits, our simulations show a classification accuracy of 82.9% (Fig. S15). These results demonstrate that the analog characteristic of FFDs has the potential for in-memory computing applications.

To further confirm the in-memory computing application of our FFD devices, we have fabricated a passive crossbar array with 1.6 k units (Fig. 6a, b). The ferroelectricity is checked by transient *I–V* curves in 200 random-selected devices where transient current peaks correspond to ferroelectric coercive voltages (Fig. S16). A uniformity with a σ/μ value of -0.023 for positive coercive voltage and -0.019 for negative coercive voltage in a Gaussian distribution is obtained (Fig. 6c). The coercive voltage and diode characteristic together enable the intended programing in the FFD passive crossbar array (see supplementary note 1, Fig. S17). The resistive switching in 400 devices at cross points of alternate rows and alternate columns is carefully checked one by one. Stirringly, all units show resistive switching (Fig. S18). These ON and OFF conductance states can be distinguished clearly (Fig. S19) and the ON/OFF ratio at the read voltage of 3 V fluctuates around 10 (Fig. 6d). We believe that the

decay of ON/OFF ratio results from the sneak path issue while the fluctuation of electrical performances is due to the immature fabrication techniques.

A simple pattern recognition task based the 1.6 k passive crossbar array is then demonstrated. Figure 6e–g display the schematic diagram for clarifying three 4 × 4-pixel images using a hardware-based artificial neural network (ANN). The images of "L", "u" and "n" are trained using the Manhattan update rule (Fig. S20) to obtain weights guiding to program the hardware ANN for recognizing the three letters. A region with 16 × 6 units are chosen to demonstrate the pattern recognition task where each weight is encoded by conductance difference between neighboring pair units (Fig. 6f, g). Fig. S21 shows the programming process and final conductance distribution in the 16 × 6 hardware ANN. During the inference progress, the content pixels are encoded into a voltage pulse with a width of 10 ms and vacant pixels into a voltage pulse with a width of 0 ms, that is absence of a voltage pulse. The patterns recognition is implemented successfully by collecting these column currents where the current difference ($I_i^+-I_i^-$) standing for the target image show much large value than others (Fig. 6h–j). Images with noise are also tested to confirm the tolerance of our hardware ANN (Fig. S22). As summarized in Fig. 6k, when zero, one, and two pixels are randomly flipped, the recognition accuracy is 100%, 50%, and 37.5%, respectively. The pattern recognition task mentioned above was automatically measured using multi-channel array test system as shown in Fig. S23.

## Comparison with state-of-the-art non-volatile memory devices

Key parameters of state-of-of-the-art non-volatile memories including Not And logic gates (NAND Flash), phase change memory (PCM), FeRAM, resistive RAM (RRAM) and Magnetic RAM (MRAM) have been collected in recent review reports[41]. A comparison of the performance with other memories is summarized in Table S1. Among the vast family of nonvolatile memories, our FFD memory cumulatively demonstrates very high performances with an endurance of over $10^{10}$ cycles, a self-rectification ratio of ~$10^4$, an operation speed of 100 ns, a feature size of 30 nm and cell size of 4 $F^2$, and an ultralow energy consumption of ~20 fJ. The simple two-terminal structure and the high self-rectification ratio of ~$10^4$ permit to efficiently design passive crossbar arrays for high-density memories as well as emerging in-memory computing application.

## Discussion

A robust ferroelectric-based non-volatile memory with a novel FFD structure is proposed as a new building block for future electronic circuit architectures. The device absorbs merits of non-destructive read mode with resistive switching as in FTJs and long endurance as in FeRAM while it allows ferroelectric directly on electrode other than semiconductor, eliminating the intrinsic source of device-to-device variation in a traditional FeFET. Both digital and analog memory functionalities can be achieved in such device. It can operate with different ferroelectric materials illustrating its universal character. It demonstrates superior performances when compared to state-of-the-art nonvolatile memories with an endurance of over $10^{10}$ cycles, an ON/OFF ratio of ~$10^2$, a feature size of 30 nm and cell size of 4 $F^2$, an operating energy of ~20 fJ and an operation speed of 100 ns. Analog storage using multiple conductance states is demonstrated showing such a device is suitable for synaptic learning plasticity. The simple two-terminal structure and its self-rectifying ratio of ~$10^4$ permit a passive crossbar array with 1.6 k units in which in-memory computing of a simple pattern classification task is accomplished. The high device-to-device uniformity is reflected by a small σ/μ value of ~0.023 for positive coercive voltage and ~0.019 for negative coercive voltage using a Gaussian distribution. This work paves a way to use this new electronic unit for designing passive crossbar arrays for either memories or in-memory computing applications.

## Methods

### P(VDF-TrFE) based FFD

The structure and fabrication process of the designed FFD are orderly presented in Fig. 2a. First of all, the striped 100-nm Pt films sputtered by DC magnetron sputtering instrument on the cleaned SiO₂/Si were used as the gate and drain electrodes. The working power, chamber pressure, and Ar flow rate were 250 W, 0.3 Pa, and 50 sccm, respectively (step 1); Secondly, the P(VDF-TrFE) (70:30 mol%) ferroelectric polymer was dissolved in the diethyl carbonate with 2.5 wt%. The P(VDF-TrFE) is formed by spin coating the polymers in an initially homogeneous evaporative solution, followed by a thermal anneal at 135 °C for 4 h to facilitate the ferroelectric β-phase growth as shown in step 2; Thirdly, the ~20-nm Al was deposited on the 120-nm ferroelectric layer by thermal evaporation (the deposition rate is 10 Å/s); The crossbar structure between top electrodes and bottom electrodes are accomplished by switching the metal mask before depositing the electrodes. Next, the excess organic polymer (P(VDF-TrFE)) was removed by oxygen ion etching, and the P(VDF-TrFE) films covered by Al was retained. Finally, assisted by a metal mask, the 30-nm ZnO was sputtered onto the crossbar network using RF magnetron sputtering instrument at a chamber pressure of 0.8 pa, Ar flow rate of 50 sccm and working power of 80 W.

### PZT based FFD

The structure and fabrication process of the designed FFD based on PZT are orderly presented in Fig. S5. Assisted by a metal mask, the ~300-nm Al electrodes with a periodic striped structure were deposited on the purchased PZT (200 nm)/Pt (50 nm)/SiO₂/Si substrate by thermal evaporation (the deposition rate is 10 Å/s). The excess PZT was removed by argon ion etching, and the PZT covered by Al electrodes was retained. Finally, assisted by a metal mask, the 30-nm ZnO was sputtered to cover the electrodes boundary using RF magnetron sputtering instrument with a chamber pressure of 0.8 pa, Ar flow rate of 50 sccm, working power of 80 W.

### The reversed FFD

The fabrication process of the designed reversed FFD based on P(VDF-TrFE) are orderly presented in Fig. S11. Firstly, assisted by lithography, the 100 nm-thick Al electrodes were deposited on the purchased SiO₂/Si substrate by thermal evaporation (the deposition rate is 10 Å/s) (step1). Secondly, the P(VDF-TrFE) (70:30 mol%) ferroelectric polymer was dissolved in the diethyl carbonate with 2.5 wt %. The P(VDF-TrFE) ( ~ 360 nm, 6 layers) is formed by spin coating the polymers in an initially homogeneous evaporative solution, followed by a thermal anneal at 135 °C for 4 h to facilitate the ferroelectric β-phase growth as shown in step 2. The Pt electrode was prepared by means of photolithographic alignment as shown in Fig. S11b–d and step 3. The Pt films are sputtered by DC magnetron sputtering instrument, and the working power, chamber pressure, and Ar flow rate were 250 W, 0.3 Pa, and 50 sccm, respectively. Thirdly, the excess P(VDF-TrFE) was removed by oxygen ion etching, and the P(VDF-TrFE) covered by Pt electrodes was retained (step 4). Finally, assisted by a metal mask, the 30-nm ZnO was sputtered to cover the electrodes boundary using RF magnetron sputtering instrument with a chamber pressure of 0.8 pa, Ar flow rate of 50 sccm, working power of 80 W (step 5).

### A FFD nano device

The fabrication process of the designed FFD nano device based on P(VDF-TrFE) are orderly presented in Fig. S12. First of all, the striped 100-nm Pt films sputtered by DC magnetron sputtering instrument on the cleaned SiO₂/Si were used as the gate and drain electrodes (step 1). The working power, chamber pressure, and Ar flow rate were 250 W, 0.3 Pa, and 50 sccm, respectively; Secondly, the P(VDF-TrFE) (70:30 mol%) ferroelectric polymer was dissolved in the diethyl

carbonate with 2.5 wt%. The 60 nm-thick P(VDF-TrFE) (~60 nm, 1 layer) is formed by spin coating polymers in an initially homogeneous evaporative solution, followed by a thermal anneal at 135 °C for 4 h to facilitate the ferroelectric $\beta$-phase growth as shown in step 2; Thirdly, assisted by lithography, the ~1-μm photoresist (PMMA) with a periodic striped structure were prepared on the P(VDF-TrFE) (60 nm)/Pt (50 nm)/SiO$_2$/Si substrate by spin coating and a thermal anneal (200 °C for 2 h) (step 3). Next, assisted by a metal mask, the 30-nm Al was deposited onto the crossbar network using thermal evaporation instrument (the deposition rate is 10 Å/s) (step 4). The excess Al was removed by argon ion etching, and the Al upright and close to the PMMA was retained as shown in step 5. Next, the remaining P(VDF-TrFE) was cleaned up by oxygen ion etching (step 6). Finally, assisted by a metal mask, the 30-nm ZnO was sputtered onto the crossbar network using RF magnetron sputtering instrument at a chamber pressure of 0.8 pa, Ar flow rate of 50 sccm and working power of 80 W (step 7).

## A 40 × 40 passive crossbar array
Both the bottom electrode and the top electrode are assisted by striped mask plates arranged with 40 columns. The detailed preparation process is the same as that in the P(VDF-TrFE)-based FFD devices.

## STEM measurements
The Pt/P(VDF-TrFE)/Al/ZnO STEM sample was fabricated using a focused ion beam machine (Hellios G4 UX). Structural characterization was conducted using a JEM- ARM200 scanning transmission electron microscope equipped with an ASCOR probe corrector operating at an accelerating voltage of 300 kV.

## Electrical measurements
The $I$–$V$ curves, programing speed and retention property of the FFD were measured, unless noted otherwise, under ambient conditions using a Keithley 4200A-SCS parameter analyzer with remote pre-amplifiers. The read voltage of 5 V (0.3 V) is used to obtain the retention characteristic of the FFD based on P(VDF-TrFE) (PZT). A HP4194A impedance analyzer with an ac amplitude of 0.04 V is used to measure the capacitance versus frequency ($C$-$f$) at various temperatures. The $P$–$V$ hysteresis loops and the fatigue characteristic of the FFD were investigated using ferroelectric analyzer (TF Analyzer 3000). In fatigue measurements, the fatigue pulses with an amplitude of 25 V (10 V) and a frequency of 10 kHz (2 MHz) were used to reverse the ferroelectric domains in P(VDF-TrFE) (PZT) based devices. The checked $P$–$V$ curves were measured using a triangular wave with the same amplitude and frequency. A quasi static $I$–$V$ sweeping was performed after ferroelectric fatigue texts with different cycles to obtain the cycles-dependent $I$–$V$ curves. For all temperature-dependent measurements, the temperature was changed at a rate of 1 K/min. Each temperature was maintained for 2 mins using a computer-controlled cryostat (MMR Tech, Inc.) before the following electrical measurements. The electrical performance of the FFD array was achieved by a multi-channel array test system (PXIe-4631) controlled by a labview program.

## TCAD simulation
Silvaco TCAD simulator has been used to simulate the device we proposed. The structure is created by a 2D process simulation editor (Athena), where the size parameters of device modeling are consistent with the experiments. The contact between Pt and ZnO was defined as a Schottky contact with a barrier height of 1.02 eV, and the contact between Al and ZnO was set as an ohmic contact by default. A doping concentration in the ZnO is set as $10^{20}$ cm$^{-3}$. The dielectric constant and the band gap of P(VDF-TrFE) are set as 10.7 and 6 eV, respectively. Physical models including mobility, recombination, and fermi-Dirac are employed in the simulation.

## Data availability
The data that support the findings of this study are available from the corresponding author upon reasonable request.

## Code availability
The codes that support the findings of this study are available from the corresponding author upon reasonable request.

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

## Acknowledgements

B.T. would like to thank fundings of National Key Research and Development Program of China (No. 2021YFA1200700), National Natural Science Foundation of China (No. T2222025, 62174053 and 61804055), Open Research Projects of Zhejiang Lab (2021MD0AB03), Shanghai Science and Technology Innovation Action Plan (No. 19JC1416700, 21JC1402000 and 21520714100) and the Fundamental Research Funds for the Central Universities.

## Author contributions

B.T. conceived the concept. B.T., Q.Z. and C.D. supervised the research. G.F. fabricated the devices. G.F., L.C. and B.T. performed the electrical and piezoelectric force microscope measurements. G.F., S.X. and Q.B. performed the ultraviolet photoelectron spectroscopy. B.T., G.F., X.Z., J.L., K.S. and J.J. performed the simulations and experimental classification tasks. Z.Y. and R.H. performed the STEM. X.L., B.T. and W.T. performed the TCAD. Q.Z., F.Y., H.P., X.T., X.G., J.W., A.J., B.D., J.C. and C.D. advised on the experiments and data analysis. G.F., B.T. and B.D. co-write the manuscript. All authors discussed the results and revised the manuscript.

## Competing interests

The authors declare no competing interests.
