## [Peer Review File · Nature Communications]

A ferroelectric fin diode for robust non-volatile memoryEditorial Note: Parts of this Peer Review File have been redacted as indicated to remove third-party material where no permission to publish could be obtained.

REVIEWER COMMENTS

Reviewer #1 (Remarks to the Author):

The manuscript thoroughly explores an innovative two-terminal vertical ferroelectric device known as the 2T-vertical FeFET, emphasizing its remarkable characteristics. This device seamlessly integrates digital and analog memory functionalities, offering impressive endurance, a compact form factor, minimal energy consumption, and swift operational capabilities. Additionally, the authors investigate its potential as a foundational element for passive crossbar arrays, a critical component in future in-memory computing applications.

However, I must express reservations regarding the classification of these devices as Field Effect Transistors (FETs). The physical consolidation of drain and gate contacts into a single layer suggests a two-electrode system, with Schottky and ohmic contacts playing pivotal roles, contributing to a diode-type configuration. Furthermore, the available evidence does not strongly support the classification of these devices as FETs.

One notable aspect requiring further clarification is how this innovative architecture effectively manages device-to-device variation while ensuring uniformity across all devices.

To enhance the manuscript's accessibility and reader comprehension, it would be immensely beneficial to include a comparative table presenting key performance metrics in comparison to state-of-the-art nonvolatile memories. Additionally, Figure 1, as currently presented, provides a schematic representation but lacks the quantitative details of key parameters.

It would be helpful to understand how the fabrication process for these 2-T vertical FeFETs compares in terms of simplicity and efficiency when contrasted with existing technologies.

Reviewer #2 (Remarks to the Author):

This paper presents a two-terminal vertical ferroelectric device which is realized by connecting the gate and drain of a three-terminal ferroelectric field effect transistor (FeFET) together. The authors integrate the device with ferroelectric P(VDF-TrFE) and PZT and explained the device operation mechanisms, reported the device performance, and demonstrated the device application for in-memory computing applications. Overall, the device integration is good, but there are a few challenges of this work.

1. The title is misleading. At the end, it is not a transistor, but two terminal device. Also the terminology of "2-T" and "3-T" is also misleading as "2-T" is typically considered as two transistor cell.

2. A key challenge of this design is that it loses the energy efficiency of ferroelectric memory. With this design, the switching of ferroelectric polarization under positive voltage is accompanied with the excessive channel current, which is not present in conventional FeFET. As a result, it brings back all the challenges of two-terminal memory devices that a selector is required for each cell such that the sneak paths are cutoff. It is also unclear how the 100fJ energy consumption is calculated.

3. The claim of self-rectifying ratio of 10^4 is also misleading. That ratio alone does not guarantee the correct operation of passive crossbar array. Since the inhibition bias scheme needs to be applied, the I_{ON}/I_{OFF} at half (or one third) of the write bias matters. Given write voltage of 20V in Fig.3a, the ratio is even less than 10. It is unclear how the authors program the passive crossbar array. To demonstrate successful array operation, the authors are encouraged to demonstrate the programmed resistance value across the array.

4. The authors claim that the two-terminal device can eliminate the intrinsic source of device-to-device variation. It is unclear where that comes from. Indeed the ferroelectric is deposited on the gate/drain electrode, but the operation mechanism introduces new variation sources. Without serious investigation, such a claim is ungrounded.

5. When describing the I-V of the device, the authors mention that "Moreover, the I-V curve of the 2-T vertical FeFET further demonstrates that the structure of this new device can significantly reduce the reverse ($V < 0$) tunneling leakage current (Fig. S1a)." Why the reverse current is tunneling based?

6. When describing the negative reset process, the authors mentioned that "The remanent polarization with net negative bound charges at the ferroelectric/channel interface electrically weakens the doping density of the n-type channel and leads to a thicker Schottky barrier (Fig. 3i), resulting in the low conductance state (LCS)." It is ambiguous what it means that doping density is weakened.

7. The design shown in Fig.4d is fundamentally different from the main concept in the paper as the Pt now gates the ZnO through the ferroelectric.

8. In the last, there needs to be a comprehensive discussion on the scalability of the design from both the lateral and vertical dimensions. In the vertical direction, the author mentioned the thickness can be reduced for voltage reduction. However, there is a tradeoff between the voltage and the conduction current and associated power consumption. In the 2-terminal design, many of the parameters are inter-related, introducing many challenges.

Point-to-point reply letter

Response to the Reviewer #1's comments

General comments: The manuscript thoroughly explores an innovative two-terminal vertical ferroelectric device known as the 2T-vertical FeFET, emphasizing its remarkable characteristics. This device seamlessly integrates digital and analog memory functionalities, offering impressive endurance, a compact form factor, minimal energy consumption, and swift operational capabilities. Additionally, the authors investigate its potential as a foundational element for passive crossbar arrays, a critical component in future in-memory computing applications.

Response: We thank and appreciate the reviewer#1 for his/her positive comments on our work. All of these comments have been well evidenced by our further experiments in the revised manuscript. Below is a point-by-point response to the questions raised by the Reviewer #1.

Comments 1-1: I must express reservations regarding the classification of these devices as Field Effect Transistors (FETs). The physical consolidation of drain and gate contacts into a single layer suggests a two-electrode system, with Schottky and ohmic contacts playing pivotal roles, contributing to a diode-type configuration. Furthermore, the available evidence does not strongly support the classification of these devices as FETs.

Response 1-1: Thank the reviewer#1 for his/her valuable comments on the name of the “two-terminal ferroelectric memory”. We agree with reviewer#1’s viewpoint that the structure is “a diode-type configuration”. In this “two-terminal ferroelectric memory”, a ferroelectric capacitor and a fin-like semiconductor channel are combined to share both top and bottom electrodes. A diode-type (Schottky) contact between the fin-like semiconductor channel and one of the electrodes is involved. While the fin-like semiconductor channel contributes most currents, the ferroelectric domain switching rearranges the electric field configuration at ferroelectric/semiconductor interface and results in a resistive switching. To avoid any potential misunderstandings, we have redefined this two-terminal ferroelectric memory as “ferroelectric fin diode”. We

changed manuscript's title to "A ferroelectric fin diode for robust non-volatile memory" and modified all device name correspondingly in the revised manuscript.

Comments 1-2: One notable aspect requiring further clarification is how this innovative architecture effectively manages device-to-device variation while ensuring uniformity across all devices.

Response 1-2: Thank the reviewer#1 for his/her concern on the device uniformity. In a traditional ferroelectric transistor (**Fig. R1a**), the ferroelectric layer has to be deposited on the semiconductor layer other than a seed electrode. The lack of an epitaxial template results in poor ferroelectric quality that is usually mesoscopically disordered and polycrystalline¹. This leads to uncontrolled device-to-device variation in nanoscale devices¹. On the contrary, in a ferroelectric fin diode (**Fig. R1b**), the ferroelectric layer is directly deposited on a seed electrode (for example, Pt electrode for PZT films) and covered by a top electrode. This sandwiching metal/ferroelectric/metal structure, indeed the same as that of a commercial ferroelectric capacitor, possesses a high ferroelectric quality and good uniformity even in nanoscale. Note that during the following semiconductor deposition, the ferroelectric layer is protected by the top electrode.

Figure R1 (a-b), The sketch of a traditional ferroelectric transistor (a) and a ferroelectric fin diode (b). To confirm the uniformity of the ferroelectric fin diodes, the device-to-device variation in a 4×4 array (**Fig. R2**) and a 40×40 array (**Fig. R3**) is checked. The device uniformity in the 4×4 array (**Fig. R2a**) is checked by performing I - V curves in all 16 device units (**Fig. R2b**) which shows that the dispersion of the response from one device unit to another is very small. The device-to-device variation is evaluated using the ratio of σ/μ in a Gaussian distribution function $f(G) = \frac{1}{\sqrt{2\pi}\sigma} e^{-\frac{(G-\mu)^2}{2\sigma^2}}$, where μ and σ are the mean value and standard deviation of the current, respectively. A good uniformity is found with a σ/μ value of ~ 0.18 for HCS and ~ 0.08 for LCS, respectively (**Fig. R2c**). The

device uniformity in the 40×40 array (**Figs. R3a-b**) is checked by measuring transient I - V curves of 200 random-selected devices where transient current peaks correspond to ferroelectric coercive voltages (**Fig. R3c**). A uniformity with a σ/μ value of ~ 0.023 for positive coercive voltage and ~ 0.019 for negative coercive voltage in a Gaussian distribution is obtained (**Fig. R3d**).

Figure R2 The uniformity of ferroelectric fin diodes in a 4×4 passive crossbar array. **(a)** The schematic diagram of a 4×4 passive crossbar array constituted by P(VDF-TrFE)-based fin diodes. Insets show the schemes (left) and the optical image (right) of a unit device. **(b)** The overlay plots of quasi-static I - V curves of all 16 devices. **(c)** The distribution of low LCS and high HCS conductance states for all 16 devices. The red lines indicate the fit by gaussian distribution.

Figure R3 The uniformity of ferroelectric fin diodes in a 40×40 passive crossbar array. **(a-b)**, The optical image **(a)** and schematic diagram **(b)** of a 40×40 passive crossbar array constituted by P(VDF-TrFE)-based fin diodes. **(c)** The transient I - V curves in 200 random-selected devices where transient current peaks correspond to ferroelectric coercive voltages. **(d)** The distribution of negative coercive voltage and positive coercive voltage obtained from **(c)**.

We added these discussions to the revised manuscript. See “In a traditional ferroelectric transistor (**Fig. 1c**), the ferroelectric layer has to be deposited on the semiconductor layer other than a seed electrode. The lack of an epitaxial template results in poor ferroelectric quality that is usually mesoscopically disordered and polycrystalline(46). This leads to uncontrolled device-to-device variation in nanoscale devices. On the contrary, in a FFD (**Fig. 1d**), the ferroelectric layer is directly deposited on a seed electrode (for example, Pt electrode for PZT films) and covered by a top electrode. This sandwiching metal-ferroelectric-metal structure, indeed the same as that of a commercial ferroelectric capacitor, possesses a high ferroelectric quality and good uniformity even in nanoscale. Note that during the following semiconductor deposition, the ferroelectric layer is protected by the top electrode. Thus, a good uniformity is expected in FFD devices.” that is highlighted by blue color in page 12.

Reference:

1. A. I. Khan, A. K., S. Datta, The future of ferroelectric field-effect transistor technology, Nat. Electron. 2020, 3, 588-597;

Comments (1-3): To enhance the manuscript's accessibility and reader comprehension, it would be immensely beneficial to include a comparative table presenting key performance metrics in comparison to state-of-the-art nonvolatile memories. Additionally, Figure 1, as currently presented, provides a schematic representation but lacks the quantitative details of key parameters.

Response 1-3: We appreciate the reviewer#1 for his/her suggestion of “a comparative table presenting key performance metrics in comparison to state-of-the-art nonvolatile memories”, which is very helpful for us to improve the quality of the article. As shown in **table R1**, key parameters of state-of-the-art non-volatile memories including Not And logic gates (NAND Flash), phase change memory (PCM), FeRAM, resistive RAM (RRAM) and Magnetic RAM (MRAM) were obtained from recent review reports¹. Among the vast family of nonvolatile memories, this new ferroelectric fin diode cumulatively demonstrates very high performances. The **table R1** is presented as **table S1** in the supplementary materials.

Table 1. Performance comparison with state-of-the-art non-volatile memories.

	NAND Flash	PCM	FeRAM	RRAM	MRAM	This work
Cell size	4/176L F ²	4/4L F ²	6 to 30 F ²	6 to 30 F ²	6 to 30 F ²	4 F ²
Energy	~ 0.01 pJ	~ 10 pJ	~ 0.1 pJ	~ 0.1 pJ	~ 0.1 pJ	~ 0.1 pJ
Speed	~ 10 μ s	10 to 100 ns	10 to 100 ns	~ 100 ns	~ 10 ns	~ 100 ns
Endurance	~ 10 ⁴	~ 10 ⁷	~ 10 ¹⁵	~ 10 ⁶	~ 10 ¹⁵	10 ¹⁰
Rectification	1	1		1	1	10 ⁴

Additionally, as shown in **Figure R4**, we have added quantitative details of these parameters in Figure 1 of the revised manuscript. The detailed parameters in FeRAM, FTJ and FeFET devices are obtained from ref 1, 2 and 3 respectively.

[REDACTED]

Figure R4 Comparison of ferroelectric memory performances. (a-d) In the ferroelectric random access memory capacitor (FeRAM) (a), Ferroelectric tunnel junctions (FTJ) (b), conventional 3-Terminal ferroelectric field-effect transistor (FeFET) (c) and the proposed ferroelectric fin diode (FFD) (d). The parameters in FeRAM, FTJ and FeFET devices are obtained from ref 1, 2 and 3 respectively.

Reference:

1. M. Lanza, A. Sebastian, W. D. Lu, M. L. Gallo, M. Chang, D. Akinwande, F. M. Puglisi, H. N. Alshareef, M. Liu, J. B. Roldan, Memristive technologies for data

- storage, computation, encryption, and radio-frequency communication, The future of ferroelectric field-effect transistor technology, Science 2022, 376, eabj9979;
- Z. Wen, D. Wu, Ferroelectric Tunnel Junctions: Modulations on the Potential Barrier, Adv. Mater. 2019, 32, 1904123;
 - A. I. Khan, A. K., S. Datta, The future of ferroelectric field-effect transistor technology, Nat. Electron. 2020, 3, 588-597;

Comments (1-4): It would be helpful to understand how the fabrication process for these 2-T vertical FeFETs compares in terms of simplicity and efficiency when contrasted with existing technologies.

Response (1-4): We thank Reviewer#1 for his/her suggestion on discussion of the fabrication process. The simplified fabrication process of the ferroelectric fin diode is presented in Figure R5. Either standard lithography technique or metal mask can be used to form the graphic electrodes. For the P(VDF-TrFE)-based ferroelectric fin diode (**Fig. R6a**), during the oxygen etching process, the Al electrode protects the P(VDF-TrFE) films below the Al electrode, forming the step (4) for the following deposition of ZnO films (6). For the PZT-based ferroelectric fin diode (**Fig. R6a**), argon ion etching technique is used to form the step (4) for the following deposition of ZnO films (6). In our work, we mostly intended to mention the simple two-terminal structure, to avoid any misunderstandings, we changed “simplicity to fabricate” and “simple fabrication process” by “simple two-terminal structure” in the revised manuscript.

Figure R5 (a-b) The fabrication process of the ferroelectric fin diodes based on P(VDF-TrFE) (a) and PZT (b).

Response to the Reviewer #2's comments

General comments: This paper presents a two-terminal vertical ferroelectric device which is realized by connecting the gate and drain of a three-terminal ferroelectric field effect transistor (FeFET) together. The authors integrate the device with ferroelectric P(VDF-TrFE) and PZT and explained the device operation mechanisms, reported the device performance, and demonstrated the device application for in-memory computing applications. Overall, the device integration is good, but there are a few challenges of this work.

Response: We thank and appreciate the reviewer#2 for his/her positive comments on our work. Below is a point-by-point response to the questions raised by the Reviewer #2.

Comments (2-1): The title is misleading. At the end, it is not a transistor, but two terminal device. Also the terminology of "2-T" and "3-T" is also misleading as "2-T" is typically considered as two transistor cell.

Response (2-1): We thank the reviewer#2 for his/her reminder of the “misleading” title. In this “two-terminal ferroelectric memory”, a ferroelectric capacitor and a fin-like semiconductor channel are combined to share both top and bottom electrodes. A diode-type (Schottky) contact between the fin-like semiconductor channel and one of the electrodes is involved. While the fin-like semiconductor channel contributes most currents, the ferroelectric domain switching rearranges the electric field configuration at ferroelectric/semiconductor interface and results in a resistive switching. To avoid any potential misunderstandings, we have redefined this two-terminal ferroelectric memory as “ferroelectric fin diode”. We changed manuscript’s title to “A ferroelectric fin diode for robust non-volatile memory” and modified all device name correspondingly in the revised manuscript.

Comments (2-2): A key challenge of this design is that it loses the energy efficiency of ferroelectric memory. With this design, the switching of ferroelectric polarization under positive voltage is accompanied with the excessive channel current, which is not present

in conventional FeFET. As a result, it brings back all the challenges of two-terminal memory devices that a selector is required for each cell such that the sneak paths are cutoff. It is also unclear how the 100fJ energy consumption is calculated.

Response (2-2): We thank the reviewer#2 for his/her concern about the energy consumption. We agree with reviewer#2 that the excessive channel current will add the energy consumption for write operations. The write speed of a P(VDF-TrFE)-based device can be of $\sim 1 \mu\text{s}$ (**Fig. R6** and **Fig. S10a** in the revised manuscript). The energy consumption for each write operation is evaluated by the formula: $E=UIt$. For example, when an operating voltage with an amplitude of 20 V is used for a reversed ferroelectric fin diode based on P(VDF-TrFE) (**Fig. R7b** and **Fig. S11f** in the revised manuscript), the energy consumption is estimated to be $\sim 1 \text{ fJ}$ for a reset operation ($-20 \text{ V} \cdot -5 \times 10^{-11} \text{ A} \cdot 10^{-6} \text{ s} = 1 \text{ fJ}$) and $\sim 20 \text{ fJ}$ for a set operation ($20 \text{ V} \cdot 10^{-9} \text{ A} \cdot 10^{-6} \text{ s} = 20 \text{ fJ}$). When an operating voltage with an amplitude of 28 V, which enables a higher ON/OFF ratio, is used for the reversed ferroelectric fin diode, the energy consumption is estimated to be $\sim 1.68 \text{ fJ}$ for a reset operation ($-28 \text{ V} \cdot -6 \times 10^{-11} \text{ A} \cdot 10^{-6} \text{ s} = 1 \text{ fJ}$) and $\sim 280 \text{ fJ}$ for a set operation ($28 \text{ V} \cdot 10^{-8} \text{ A} \cdot 10^{-6} \text{ s} = 280 \text{ fJ}$). It should be noted that the currents in ferroelectric fin diodes are self-rectified, it means that only forward currents are allowed if they are involved in a crossbar array. As demonstrated in **Fig. R8**, the diode characteristic indeed effectively inhibits the sneak currents¹.

We have added these discussions to the revised manuscript. See “The energy consumption for each write operation of the ferroelectric fin diode is evaluated using the formula: $E=UIt$. For example, when an operating voltage with an amplitude of 20 V is used for a reversed FFD based on P(VDF-TrFE) (**Fig. S11f**), the energy consumption is estimated to be $\sim 1 \text{ fJ}$ and $\sim 20 \text{ fJ}$ for a reset operation and a set operation respectively.” that is highlighted by blue color in page 12.

Figure R6 The programed voltage pulse sequence (top panel) and the evolution of conductance with time (bottom panel) for a typical ferroelectric fin diode based on P(VDF-TrFE).

Figure R7 (a) The sketch of a reversed ferroelectric fin diode based on P(VDF-TrFE). (b) The multiple quasi-static I - V curves of the reversed ferroelectric fin diode with a sweeping amplitude of 16 V, 20 V, 24 V and 28 V, respectively.

Figure R8 (a-b) A diagram (a) and equivalent circuit (b) of a 2×2 crossbar array containing memory elements and diodes in series, in which the sneak path current is inhibited. The figure is obtained from

Ref. 1.

Reference:

1. L. Shi, G. Zheng, B. Tian, B. Dkhil, C. Duan, Research progress on solutions to the sneak path issue in memristor crossbar arrays. *Nanoscale Adv.* 2020, 2, 1811–1827;

Comments (2-3): The claim of self-rectifying ratio of 10^4 is also misleading. That ratio alone does not guarantee the correct operation of passive crossbar array. Since the inhibition bias scheme needs to be applied, the I_{ON}/I_{OFF} at half (or one third) of the write bias matters. Given write voltage of 20V in Fig.3a, the ratio is even less than 10. It is unclear how the authors program the passive crossbar array. To demonstrate successful array operation, the authors are encouraged to demonstrate the programmed resistance value across the array.

Response (2-3): We thank the reviewer#2 for his/her concern about the programming of the ferroelectric fin diode passive crossbar array. Indeed, a passive array architecture is appealing in for high packing density, but it suffers the sneak path current issue. The self-rectifying characteristic guarantees that only forward paths are allowed in the ferroelectric fin diode passive array which will effectively inhibit the sneak paths (**Fig. R8**)¹. It means that the target's conductance state can be "correctly" read out.

A 2×2 ferroelectric fin diode passive crossbar array is used to demonstrate the writing operations (**Fig. R9**). Under an external forward voltage, the ferroelectric fin diode can be treated as a resistance. Under an external backward voltage, the ferroelectric fin diode can be treated as a single ferroelectric capacitor because the backward conductance is extremely low. To illuminate the ferroelectric domain switching of a targeted ferroelectric fin diode, it is treated as shunt-wound ferroelectric capacitor and resistance for a targeted unit and a single resistance for untargeted units under external forward voltages.

During a set operation (**Fig. R9a**), in the red sneak path, the device 2 (D2) and device 3 (D3) are forward biased while the device 4 (D4) is backward biased. D4 suffers most part of the external voltage since its huge resistance value under the backward voltage. The equivalent circuit can be simplified as **Fig. R9b** and **Fig. R9c** for the targeted D1

and the high voltage-biased D4 respectively. Note that two resistances ($2R$), D2 and D3 under a forward voltage bias, are connected to the ferroelectric capacitor (D4) in series (**Fig. R9c**). The series resistance plays a crucial role for the domain switching process since the domain-switching speed at V is limited by the maximum current flow through R_s in the circuit, where the V and R_s is the amplitude of the external voltage and series resistance respectively². Considering that the $R_s = r$ in the target D1 circuit is more than 5 orders smaller than the $R_s = 2R+r$ in the D4 circuit, the quick programming process for the target D1 effect little on the domain configuration in D4.

During a reset operation (**Fig. R9d**), in the red sneak path, the D2 and D3 are backward biased while the D4 is forward biased. D2 and D3 together suffer most part of the external voltage since their huge resistance value under the backward voltage. If it satisfies that: $V_V/2 < V_c < V_t$, where the V_t and V_c are the external voltage and coercive voltage of the ferroelectric fin diodes respectively, the V_t can only program the domain configuration in the targeted D1.

In summary, the diode characteristic and coercive voltage together enable the intended programming in the ferroelectric fin diode passive crossbar array. The programmed ON (**Fig. R10a**) and OFF (**Fig. R10b**) conductance states, after triangular voltage wave of ± 15 V, in 400 devices at cross points of alternate rows and alternate columns of the ferroelectric fin diode passive crossbar array are carefully checked one by one using a reading voltage of 3 V. These ON and OFF conductance states can be distinguished clearly with an ON/OFF ratio of ~ 10 .

The detailed programming process in the 16×6 hardware ANN is presented in **Fig. R11a** (**Fig. S21a** in the revised supplementary materials). The conductance states are programmed column by column. And to avoid the potential error operations, the set (odd steps) and reset (even steps) operations are performed successively (**Fig. R11a**, **Fig. S21a** in the revised supplementary materials). **Figure R11b** (**Fig. S21b** in the revised supplementary materials) shows the final conductance distribution in the 16×6 hardware ANN.

Figure R9 The equivalent circuit for writing operations. (a-c), The equivalent circuit in a set operation for D1 (b) and D4 (c) from a 2×2 passive crossbar array (a). (a-c), The equivalent circuit in a reset operation for D1 (e) and D2 and D3 (f) from a 2×2 passive crossbar array (d).

Figure R10 (a-b), The programmed ON (a) and OFF (b) conductance states in 400 devices at cross points of alternate rows and alternate columns of the ferroelectric fin diode passive crossbar array.

Figure R11 (a-b) The programming process (a) and final conductance distribution (b) in the 16×6

hardware ANN. The half write voltage ($V_w/2$) is 6 V, the width of the write voltage pulse is 10 ms. The conductance distribution is checked using voltage pulses (1 V, 10 ms).

We added these discussions to the revised manuscript. See supplementary not 1 that is highlighted by blue color and **Fig. S17** and **Fig. S19** in the revised supplementary materials.

Reference:

1. L. Shi, G. Zheng, B. Tian, B. Dkhil, C. Duan, Research progress on solutions to the sneak path issue in memristor crossbar arrays. *Nanoscale Adv.* 2020, 2, 1811–1827;
2. A. Jiang, H. J. Lee, C. S. Hwang, J. F. Scott, Sub-Picosecond Processes of Ferroelectric Domain Switching from Field and Temperature Experiments, *Adv. Funct. Mater.* 2012, 22, 192–199;

Comments (2-4): The authors claim that the two-terminal device can eliminating the intrinsic source of device-to-device variation. It is unclear where that comes from. Indeed the ferroelectric is deposited on the gate/drain electrode, but the operation mechanism introduces new variation sources. Without serious investigation, such a claim is ungrounded.

Response (2-4): Thank the reviewer#2 for his/her concern on the device uniformity. In a traditional ferroelectric transistor (**Fig. R12a**), the ferroelectric layer has to be deposited on the semiconductor layer other than a seed electrode. The lack of an epitaxial template results in poor ferroelectric quality that is usually mesoscopically disordered and polycrystalline¹. This leads to uncontrolled device-to-device variation in nanoscale devices¹. On the contrary, in a ferroelectric fin diode (**Fig. R12b**), the ferroelectric layer is directly deposited on a seed electrode (for example, Pt electrode for PZT films) and covered by a top electrode. This sandwiching MFM structure, indeed the same as that of a commercial ferroelectric capacitor, possesses a high ferroelectric quality and good uniformity even in nanoscale. Note that during the following semiconductor deposition, the ferroelectric layer is protected by the top electrode.

Figure R12 (a-b), The sketch of a traditional ferroelectric transistor (a) and a ferroelectric fin diode (b). To confirm the uniformity of the ferroelectric fin diodes, the device-to-device variation in a 4×4 array (Fig. R13) and a 40×40 array (Fig. R14) is checked. The device uniformity in the 4×4 array (Fig. R13a) is checked by performing I - V curves in all 16 device units (Fig. R13b) which shows that the dispersion of the response from one device unit to another is very small. The device-to-device variation is evaluated using the ratio of σ/μ in a Gaussian distribution function $f(G) = \frac{1}{\sqrt{2\pi}\sigma} e^{-\frac{(G-\mu)^2}{2\sigma^2}}$, where μ and σ are the mean value and standard deviation of the current, respectively. A good uniformity is found with a σ/μ value of ~ 0.18 for HCS and ~ 0.08 for LCS, respectively (Fig. R13c). The device uniformity in the 40×40 array (Figs. R14a-b) is checked by measuring transient I - V curves of 200 random-selected devices where transient current peaks correspond to ferroelectric coercive voltages (Fig. R14c). A uniformity with a σ/μ value of ~ 0.023 for positive coercive voltage and ~ 0.019 for negative coercive voltage in a Gaussian distribution is obtained (Fig. R14d).

Figure R12 The uniformity of ferroelectric fin diodes in a 4×4 passive crossbar array. (a) The schematic diagram of a 4×4 passive crossbar array constituted by P(VDF-TrFE)-based fin diodes. Insets show the schemes (left) and the optical image (right) of a unit device. (b) The overlay plots of quasi-static I - V curves of all 16 devices. (c) The distribution of low LCS and high HCS conductance states for all 16 devices. The red lines indicate the fit by gaussian distribution.

Figure R13 The uniformity of ferroelectric fin diodes in a 40×40 passive crossbar array. **(a-b)**, The optical image **(a)** and schematic diagram **(b)** of a 40×40 passive crossbar array constituted by P(VDF-TrFE)-based fin diodes. **(c)** The transient I - V curves in 200 random-selected devices where transient current peaks correspond to ferroelectric coercive voltages. **(d)** The distribution of negative coercive voltage and positive coercive voltage obtained from **(c)**.

We add these discussions to the revised manuscript. See “**In a traditional ferroelectric transistor (Fig. 1c)**, the ferroelectric layer has to be deposited on the semiconductor layer other than a seed electrode. The lack of an epitaxial template results in poor ferroelectric quality that is usually mesoscopically disordered and polycrystalline(46). This leads to uncontrolled device-to-device variation in nanoscale devices. On the contrary, in a FFD **(Fig. 1d)**, the ferroelectric layer is directly deposited on a seed electrode (for example, Pt electrode for PZT films) and covered by a top electrode. This sandwiching metal-ferroelectric-metal structure, indeed the same as that of a commercial ferroelectric capacitor, possesses a high ferroelectric quality and good uniformity even in nanoscale. Note that during the following semiconductor deposition, the ferroelectric layer is protected by the top electrode. Thus, a good uniformity is expected in ferroelectric fin diodes.” that is highlighted by blue color in page 12.

Reference:

1. A. I. Khan, A. K., S. Datta, The future of ferroelectric field-effect transistor technology, Nat. Electron. 2020, 3, 588-597;

Comments (2-5): When describing the I - V of the device, the authors mention that "Moreover, the I - V curve of the 2-T vertical FeFET further demonstrates that the structure of this new device can significantly reduce the reverse ($V < 0$) tunneling leakage current (Fig. S1a)." Why the reverse current is tunneling based?

Response (2-5): We thank the reviewer#2 for his/her kind reminder on the opinionated description of the "tunneling leakage current". Here we want to mention that the reverse current is extremely small. To avoid any potential misunderstandings, we remove "tunneling leakage" away from the expression.

Comments (2-6): When describing the negative reset process, the authors mentioned that "The remanent polarization with net negative bound charges at the ferroelectric/channel interface electrically weakens the doping density of the n-type channel and leads to a thicker Schottky barrier (Fig. 3i), resulting in the low conductance state (LCS)." It is ambiguous what it means that doping density is weakened.

Response (2-6): We thank the reviewer#2 for his/her concern on the doping density by local electric field from bound charges. The doping concentration reduction here refers to the change of carrier concentration in the semiconductor due to static electricity. In our previous work^{1,2}, we have confirmed that the carrier density (Fermi level) can be effectively tuned by the ferroelectric domain configuration (**Fig. R15**). To avoid any potential misunderstandings, we replaced "doping density" by "carrier density" in the revised manuscript.

Figure R15 (a-d) Spatially defined doping in MoTe₂ using a piezoresponse force microscope-controlled ferroelectric field. (e) Output characteristics with the corresponding band diagram under different ferroelectric polarization configurations. a-d and e are obtained from Ref. 1 and Ref. 2 respectively.

Reference:

1. G. Wu et al., Programmable transition metal dichalcogenide homojunctions controlled by nonvolatile ferroelectric domains. Nat. Electron. 2020, 3, 43-50;
2. G. Wu, et al., Ferroelectric-defined reconfigurable homojunctions for in-memory sensing and computing, Nat. Mater. 2023, doi: s41563-023-01676-0;

Comments (7): The design shown in Fig.4d is fundamentally different from the main concept in the paper as the Pt now gates the ZnO through the ferroelectric.

Response: We appreciate the reviewer#2 for his/her concern on the reversed device structure. We are sorry that the reversed device structure in the old manuscript is misleading. **Figure R16** give the direct comparison between the normal device structure (**Fig. R16a**) and the reversed device structure (**Fig. R16b**). The I, II and III refer to the positive, zero and negative vertical overlap between Al electrode and Pt electrode. Note that the Schottky contact between Pt electrode and ZnO channel plays the dominated role of the resistive switching in the device. That is, the Pt electrode plays the role of “gate” for both normal device structure and reversed device structure. We added the device sketch of I, II and III to the revised manuscript. See **Fig. S11 (Fig. R17)** in the revised supplementary materials.

Figure R15 (a-b) Device sketch of a normal device structure (a) and reversed device structures (b). The I, II and III refer to the positive, zero and negative vertical overlap between Al electrode and Pt electrode respectively.

Figure R17 (a) The fabrication process of a reversed ferroelectric fin diode where the Pt electrode is on top of P(VDF-TrFE). (b) A mask with aligned patterns is used at stage (1) in (a) to photoetching Al bottom electrodes. (c) A mask with malposed patterns is used at stage (3) in (a) for photoetching Pt top electrodes. (d) The overlapped (I), “zero” overlapped (II) and separated (III) electrode pairs of Al and Pt are formed simultaneously on a same sample substrate. (e) The optical images and device sketch of

reversed devices with the overlapped (I), just “zero” overlapped (II) and separated (III) electrode pairs of Al and Pt. (f) The I - V curves of reversed ferroelectric fin diode with overlapped (I) electrode pairs of Al and Pt. During the electrical measurements, the Al electrode is always grounded. To avoid destroy of ferroelectricity during the spurring deposition of Pt electrode, thick P(VDF-TrFE) films with six spin-coating layers (~ 360 nm) are used in these reversed devices.

Comments (8): In the last, there need to be a comprehensive discussion on the scalability of the design from both the lateral and vertical dimensions. In the vertical direction, the author mentioned the thickness can be reduced for voltage reduction. However, there is a tradeoff between the voltage and the conduction current and associated power consumption. In the 2-terminal design, many of the parameters are inter-related, introducing many challenges.

Response: We thank the reviewer#2 for his/her suggestion of a discussion on the scalability. We agree with the reviewer#2 that many of the parameters in the 2-terminal resistive devices are inter-related. For example, a tradeoff issue between operation voltage and conduction current, that is, a thinner film for lower operation voltage results in higher current, is usually suffered in most 2-terminal resistive devices. In this view, it will be more energy efficient to decrease the operation voltage by using ferroelectric materials that possesses a much small coercive field. We have added these discussions to the revised manuscript. See “Note that a tradeoff issue between operation voltage and conduction current, that is, a thinner film for lower operation voltage results in higher current, is usually suffered in 2-terminal resistive devices. In this view, it will be more energy efficient to decrease the operation voltage by using ferroelectric materials that possesses a much small coercive field.” that is highlighted by blue color in page 8.

REVIEWERS' COMMENTS

Reviewer #1 (Remarks to the Author):

The authors conscientiously addressed all queries, comments, and suggestions presented in my review. Consequently, I feel quite comfortable recommending the manuscript for publication in Nature Communications.

Point-to-point reply letter

Response to the Reviewer #1's comments

General comments: The authors conscientiously addressed all queries, comments, and suggestions presented in my review. Consequently, I feel quite comfortable recommending the manuscript for publication in Nature Communications.

Response: We thank and appreciate the reviewer#1 for his/her positive comments that “The authors conscientiously addressed all queries, comments, and suggestions presented in my review. Consequently, I feel quite comfortable recommending the manuscript for publication in Nature Communications”.